

# Evolution of *Philodendron* (Araceae) species in Neotropical biomes

Leticia Loss-Oliveira[1], Cassia Sakuragui[2], Maria de Lourdes Soares[3] and Carlos G. Schrago[1]

[1] Department of Genetics, Federal University of Rio de Janeiro, Rio de Janeiro, Brazil
[2] Department of Botany, Federal University of Rio de Janeiro, Rio de Janeiro, Brazil
[3] Instituto Nacional de Pesquisas da Amazônia, Manaus, AM, Brazil

## ABSTRACT

*Philodendron* is the second most diverse genus of the Araceae, a tropical monocot family with significant morphological diversity along its wide geographic distribution in the Neotropics. Although evolutionary studies of *Philodendron* were conducted in recent years, the phylogenetic relationship among its species remains unclear. Additionally, analyses conducted to date suggested the inclusion of all American representatives of a closely-related genus, *Homalomena*, within the *Philodendron* clade. A thorough evaluation of the phylogeny and timescale of these lineages is thus necessary to elucidate the tempo and mode of evolution of this large Neotropical genus and to unveil the biogeographic history of *Philodendron* evolution along the Amazonian and Atlantic rainforests as well as open dry forests of South America. To this end, we have estimated the molecular phylogeny for 68 *Philodendron* species, which consists of the largest sampling assembled to date aiming the study of the evolutionary affinities. We have also performed ancestral reconstruction of species distribution along biomes. Finally, we contrasted these results with the inferred timescale of *Philodendron* and *Homalomena* lineage diversification. Our estimates indicate that American *Homalomena* is the sister clade to *Philodendron*. The early diversification of *Philodendron* took place in the Amazon forest from Early to Middle Miocene, followed by colonization of the Atlantic forest and the savanna-like landscapes, respectively. Based on the age of the last common ancestor of *Philodendron*, the species of this genus diversified by rapid radiations, leading to its wide extant distribution in the Neotropical region.

## INTRODUCTION

*Philodendron* is an exclusively Neotropical genus, comprising 482 formally recognized species (*Boyce & Croat, 2013*). Their geographic distribution range from Northern Mexico to Southern Uruguay (*Mayo, Bogner & Boyce, 1997*), consisting mainly of the biomes of the Amazonian and Atlantic rainforests and also the open dry forests of South America. According to *Olson et al.*'s (*2001*) classification of terrestrial biomes, South American open dry forests are composed of the Cerrado (savanna-like landscapes) and Caatinga biomes (*Croat, 1997*; *Mayo, 1988*; *Mayo, 1989*; *Coelho et al., 2016*) (Fig. 1). *Philodendron* species richness is especially significant in Brazil, where 168 species were described thus far (*Coelho et al., 2016*).

Corresponding author
Carlos G. Schrago,
carlos.schrago@gmail.com

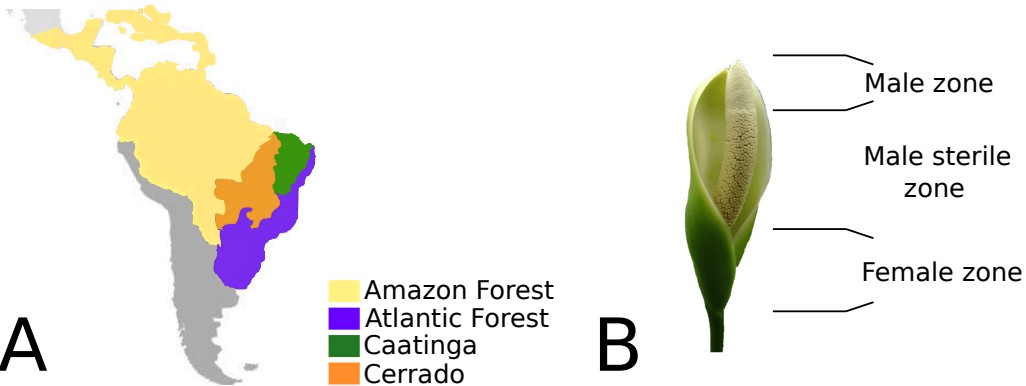

**Figure 1** (A) Geographic distribution of *Philodendron* species along the Neotropical biomes of Amazon, Atlantic forest, Cerrado and Caatinga. (B) *Philodendron* inflorescence and the flower zones.

Although *Philodendron* presents a significant morphological plasticity, wide leaf variation and several types of habits (*Coelho et al., 2016*; *Coelho, 2000*), the inflorescence morphology of its representatives is largely conserved. The unisexual flowers in the spadix are clustered in male, female and sterile zones; located at the basal, median and superior portions, respectively (Fig. 1B). The spadix, in nearly all of its extension, is surrounded by the spate (*Sakuragui, 2001*).

Currently, *Philodendron* species are grouped into three subgenera according to its floral and vegetative morphology and anatomy (*Mayo, 1991*; *Mayo, 1988*; *Croat, 1997*), namely, subgenus *Meconostigma* (Schott) Engl., which consists of 21 species (*Gonçalves & Salviani, 2002*; *Croat, Mayo & Boss, 2002*; *Mayo, 1991*); subgenus *Pteromischum* (Schott) Mayo, with 75 species (*Coelho, 2000*) and subgenus *Philodendron* (*Mayo, 1986*), comprising approximately 400 species (*Coelho, 2000*; *Croat, 1997*).

Because of the wide geographic range, patterns of distribution along niches, as well as the characteristic morphology, interest in investigating *Philodendron* systematics and evolution has increased in the last decades (*Sakuragui, Mayo & Zappi, 2005*; *Mayo, 1986*; *Grayum, 1996*; *Croat, 1997*). Morphological and anatomical characters of flowers has been of special interest for phylogenetic analysis due to their high level of variability (*Sakuragui, 1998*). However, the plasticity and convergence of these characters in *Philodendron* may increase the probability of homoplasies (*Mayo, 1986*; *Mayo, 1989*).

Recently, *Gauthier, Barabé & Bruneau (2008)* investigated the phylogenetic relationships of *Philodendron* species based on three molecular markers, sampling a total of 49 species. This work comprised the largest taxon sampling of the genus to date. In accordance to previous analysis (*Barabé et al., 2002*; *Mayo, Bogner & Boyce, 1997*), authors questioned the monophyly of *Philodendron*, suggesting the inclusion of all American species of the morphologically similar genus, *Homalomena* Schott, within the *Philodendron* clade. *Homalomena* species occur in America and Asia and its geographic distribution partly overlaps with *Philodendron* in the Neotropics. The inference of the evolutionary relationships between *Philodendron* and *Homalomena* has a significant biogeographic appeal. If American *Homalomena* species are indeed more closely related to *Philodendron*

than to Asian *Homalomena*, a single colonization event should be considered. Unveiling the evolutionary relationships between those lineages is thus necessary to elucidate their origin and subsequent diversification.

Besides phylogeny, several issues regarding *Philodendron* evolution remain unclear. For example, the historical events that led to the wide geographic occurrence along biomes need a thorough analysis. In this sense, investigating the evolutionary affinities of a large sample of *Philodendron* species will shed light on how this lineage diversified along the Amazonian and Atlantic rainforests, as well as South American open dry forests biomes; namely, the Cerrado and Caatinga. To this end, we have performed an ancestral area reconstruction of *Philodendron* and *Homalomena* species and estimated the divergence times from a phylogeny inferred from the largest *Philodendron* dataset composed to date. We were able to address the timing and pattern of *Philodendron* diversification in selected Neotropical biomes with a focus on the evolutionary relationships between the three *Philodendron* subgenera.

## MATERIALS AND METHODS

### Taxon and gene sampling

We have sequenced new data for 110 extant species of *Philodendron* and 16 species of *Homalomena* of the following molecular markers: the nuclear 18S and external transcribed spacer (ETS); and the chloroplast *trn*L intron, *trn*L-*trn*F intergenic spacer, the *trn*K intron and maturase K (*matK*) genes. Additionally, 13 outgroup species were analyzed, comprising the genera *Cercestis*, *Culcasia*, *Colocasia*, *Dieffenbachia*, *Heteropsis*, *Montrichardia*, *Nephthytis*, *Furtadoa* and *Urospatha*. Outgroup choice was based on the close evolutionary affinity of these genera to *Philodendron*, as suggested by previous studies. The complete list of species included in this study, the voucher and GenBank accession numbers were listed in Tables 1 and 2 of the Supplemental Information 1.

Ancestral biome reconstruction is dependent on the estimated phylogeny and the current geographic distribution of sampled species terminals. Thus, taxon sampling may impact the inference of ancestral species distribution along biomes. As indicated in Table S1 we have sampled all *P*. subg. *Meconostigma* species; 82 *P*. subg. *Philodendron* species and 7 *P*. subg. *Pteromischum* species. Our sampling strategy is representative of the current *Philodendron* diversity. Although ∼75% of the sampled species are *P*. subg. *Philodendron* in our analysis, ∼82% of *Philodendron* species consist of *P*. subg. *Philodendron* (*Boyce & Croat, 2013*; *Coelho et al., 2016*).

### DNA isolation, amplification and sequencing

Genomic DNA was isolated with QIAGEN DNeasy Blood & Tissue kit from silica-dried or fresh leaves. Primers used for amplification and sequencing were listed in Table S3. Sequencing reactions were performed in the Applied Biosystems 3730xl automatic sequencer and edited with the Geneious 5.5.3 software.

### Alignment and phylogenetic analysis

Molecular markers were individually aligned in MAFFT 7 (*Katoh & Standley, 2013*) and then manually adjusted in SeaView 4 (*Gouy, Guindon & Gascuel, 2010*). We estimated

individual gene trees (Fig. 1, SM) for each molecular marker in MrBayes 3.2.2 (*Huelsenbeck & Ronquist, 2001*; *Ronquist & Huelsenbeck, 2003*) using the GTR + G substitution model. The Markov chain Monte Carlo (MCMC) algorithm was ran twice for 10,000,000 generations, using four chains. Chains were sampled every 100th cycle and a burn-in of 20% was applied. A supertree was estimated from the tree topologies of each molecular marker using the PhySIC_IST algorithm, available at the ATGC-Montpellier online server (http://www.atgc-montpellier.fr/physic_ist/). Only clades with posterior probability ≥ 85% were considered to estimate the supertree. We have used this approach to avoid the impact of missing data in phylogeny estimation (*Scornavacca et al., 2008*). As PhySIC_IST calculates non-plenary supertrees, it removes taxa with significant topological conflict and/or with small taxon sampling (*Scornavacca et al., 2008*). The final supertree was thus composed of 89 terminals, as 50 terminals were discarded due to conflicting resolutions.

In order to assess the stability of the (*Philodendron* + American *Homalomena*) clade, we have calculated the log-likelihoods of alternative topological arrangements in PhyML 3.0 (*Guindon et al., 2009*) using the species sampling of the supertree. We have tested the following topologies: (I) (American *Homalomena* (*P*. subg. *Philodendron* +*P*. subg. *Meconostigma*); (II) (*P*. subg. *Meconostigma* (*P*. subg. *Philodendron* + American *Homalomena*) and (III) (*P*. subg. *Philodendron* (*P*. subg. *Meconostigma* + American *Homalomena*). The significance of the difference in log-likelihoods between topologies was tested with the approximately unbiased (AU) and the Shimodaira-Hasegawa (SH) tests implemented in CONSEL 1.2.0 (*Shimodaira & Hasegawa, 2001*).

## Divergence time inference

Dating *Philodendron* evolutionary history is difficult mainly because of the scarcity of the fossil record (*Loss-Oliveira et al., 2014*). For instance, *Dilcher & Daghlian (1977)*, based on fossilized leaves, described a putative *P*. subg. *Meconostigma* fossil from the Eocene of Tennessee (56.0–33.9 Ma). However, *Mayo (1991)* identified the referred fossil as a *Peltranda*. Thus, we have decided not to use this fossil as calibration information. Alternatively, in order to estimate divergence times, we have assigned a prior on the rate of nucleotide substitution. We were then prompted to infer the evolutionary rates of plastid coding regions of monocots using a large sample of publicly available chloroplast genomes. Nuclear genes were excluded from dating analysis because of the absence of prior information on evolutionary rates.

To estimate monocots substitution rate, we used chloroplast genomes from 154 Liliopsida species retrieved from the GenBank (Table S4). All orthologous coding regions were concatenated into a single supermatrix. Maximum likelihood phylogentic reconstruction was implemented in RaxML 7.0.3 (*Stamatakis, 2006*) under GTR model. Molecular dating of monocots (Liliopsida) was conducted under a Bayesian framework, using fossil information obtained from *Iles et al. (2015)* (Table S5). Because the number of terminals used was large, rate estimation was conducted with the MCMCTree program of PAML 4.8 package (*Yang, 2007*) using the approximate likelihood calculation (*Dos Reis & Yang, 2011*) and the uncorrelated model of evolution of rates. In MCMCTree, posterior distributions were obtained via MCMC; chains were sampled every 500th cycle

until 50,000 trees were collected. We performed two independent replicates to check for convergence of the estimates. Calibration information for Liliopsida was entered as minimum and maximum bounds of uniform priors. The estimated mean substitution rate was inferred at $3.26 \times 10^{-9}$ substitutions/site/year (s/s/y). This value is significantly higher than the previous estimate of *Palmer (1991)*, which reported an average substitution rate of $0.7 \times 10^{-9}$ s/s/y for angiosperm platids. As the credibility interval of our estimate was large, we adopted a Gaussian prior for evolutionary rates with a 95% highest probability density (HPD) interval that included maximum and minimum values of our estimate and that of Palmer's.

Dating analysis of *Philodendron* and *Homalomena* species was performed in BEAST using a relaxed molecular clock with evolutionary rates modeled by an uncorrelated lognormal distribution; the GTR+G☐model of sequence was applied. The MCMC algorithm was ran for 50,000,000 generations and sampled every 1,000th cycle, with a burn-in of 20%.

### Biome shifts

To unveil how *Philodendron* species colonized the Amazon forest, Atlantic forest, Cerrado and Caatinga, we conducted a Bayesian Binary MCMC (BBM) (*Yu, Harris & He, 2012*; *Ronquist & Huelsenbeck, 2003*) implemented in Reconstruct Ancestral State in Phylogenies 2.1b (RASP) software (*Yu, Harris & He, 2012*). The input tree topology was the supertree estimated in PhySIC_IST. BBM chains were ran for 10,000,000 generations and were sampled every 1,000th cycle. State frequencies were estimated under the F81 model with gamma rate variation. Information on the occurrence of each *Philodendron* species along Neotropical biomes was obtained from *Coelho et al. (2016)* and from the (*Team*) *CATE Araceae* (http://araceae.e-monocot.org).

## RESULTS

The *Homalomena* genus was not recovered as monophyletic; the Asian *Homalomena* clustered within a single group and the American representatives clustered independently, as sister to *Philodendron* species (Fig. 2). Although our analysis failed to support the monophyly of *Philodendron* with significant statistical support, the topological arrangement in which *Philodendron* is a monophyletic genus was significantly supported by the AU and SH tests ($p < 0.05$, Fig. 3, Table 6S). Within *Philodendron*, subg. *Meconostigma* was recovered as monophyletic (Fig. 2, node D), whereas subg. *Philodendron* was recovered as polyphyletic (Fig. 2, node E). Finally, the monophyly of *P*. subg. *Pteromischum* was not inferred, because *Pteromischum* species clustered with *P*. subg. *Philodendron* species.

We estimated that the last common ancestor (LCA) of *Philodendron* diversified in the Amazon forest (Fig. 4, node B) at ca. 8.6 Ma (6.8–12.1 Ma) 95% HPD. Thus, we inferred that the LCA of *Philodendon* diversified from Middle to Late Miocene. This also suggests that the divergence between *Philodendron* and the American *Homalomena* occurred in a short period of time after this American lineage diverged from the Asian *Homalomena* (Fig. 4, nodes B and A, respectively).

The earliest events of *Philodendron* diversification occurred exclusively in the Amazon forest (e.g., Fig. 4, nodes C, D, E, F). The ancestors of Atlantic forest lineages were inferred

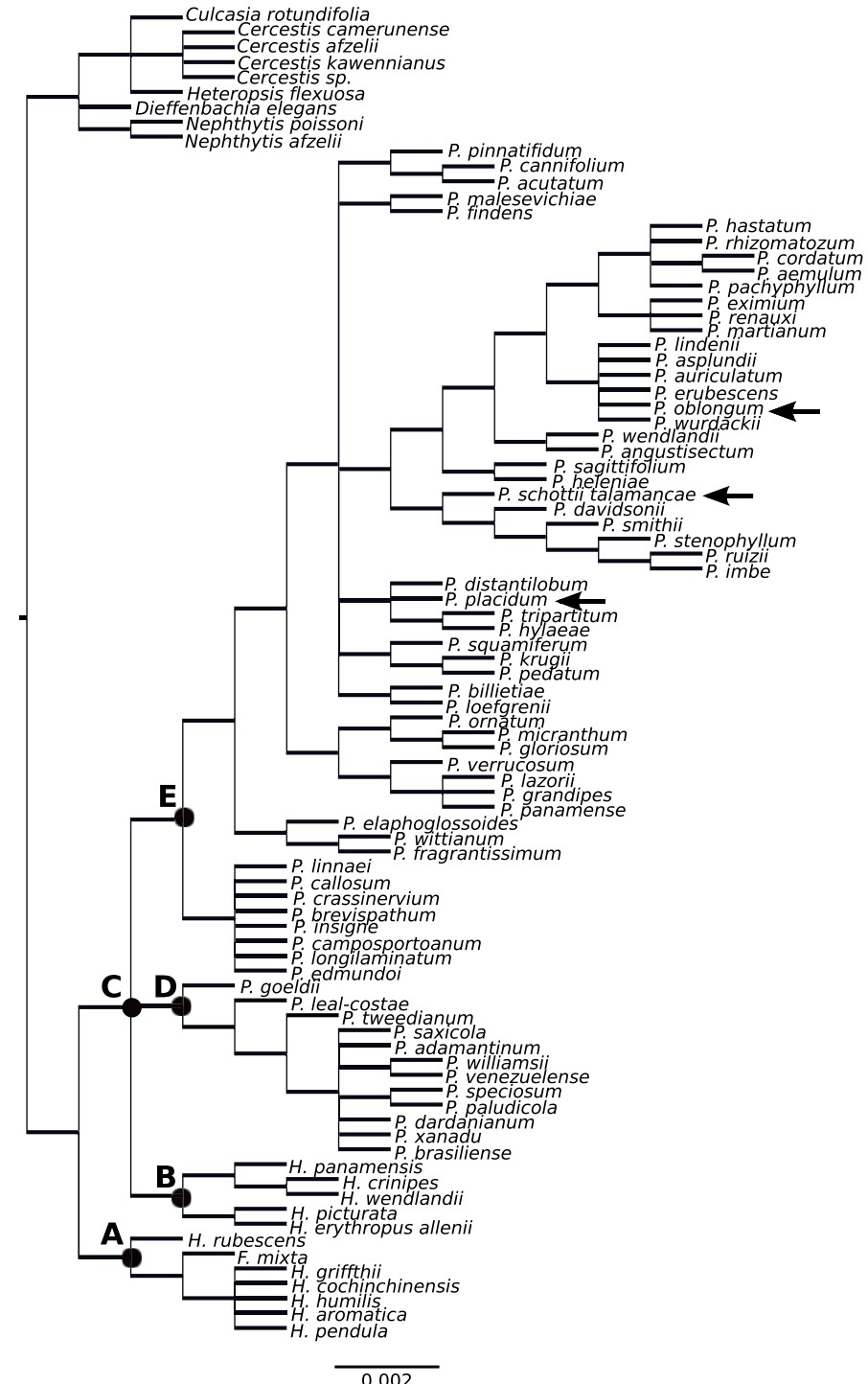

**Figure 2** Supertree of *Philodendron* and *Homalomena* species.

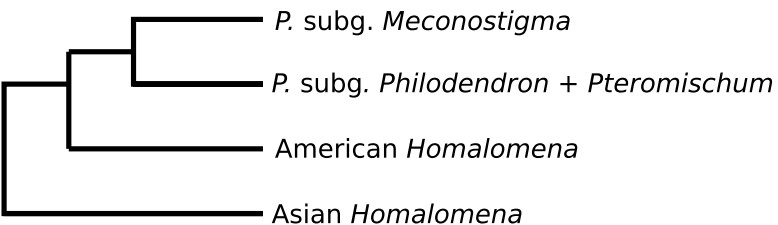

**Figure 3** Phylogeny of *Philodendron* and *Homalomena* corroborated by the approximately unbiased (AU) test.

to have been distributed in the Amazon (Fig. 4, nodes I, J and nodes G, H). This pattern of Amazonian ancestry of Atlantic forest lineages was also observed in some terminal branches. For instance, from node K to *P. loefgrenii* and from node L to *P. imbe*.

On the other hand, the majority of Cerrado species evolved from Atlantic forest ancestors (Fig. 4, nodes J and M; node N to *P. rhizomatosum* and *P. pachyphyllum*). In subgenus *Meconostigma*, the age of early species diversification into Atlantic forest was dated at 3.7 Ma (5.6–2.7 Ma) (Fig. 4, node J), whereas in the *P.* subg. *Philodendron* early lineage diversification occurred at 4.1 Ma (5.5–3.0 Ma) (Fig. 4, node J). Therefore, during a period of 5.0–6.0 Ma, *Philodendron* species occupied exclusively the Amazon forest. The diversification into Cerrado biome occurred later, at approximately 1.7 Ma (3.3–1.1 Ma) (Fig. 4, node M).

## DISCUSSION

### Phylogenetic relationship between *Philodendron* and *Homalomena*

In this study, Asian *Homalomena* was recovered as sister to the (*Philodendron* + American *Homalomena*) clade, and *Furtadoa mixta* clustered with the Asian *Homalomena* clade. The evolutionary affinities of American *Homalomena*, *P.* subg. *Meconostigma* and *P.* subg. *Philodendron* were not strongly supported. However, the topological arrangement in which *Philodendron* is a monophyletic genus was statistically significant by the AU and SH tests, suggesting the monophyly of *Philodendron*.

Previous studies have reported conflicting results concerning the monophyly of *Philodendron* and the phylogenetic status of American *Homalomena* (Fig. 5). For instance, *Barabé et al. (2002)*, based on the *trn*L intron and the *trn*L-*trn*F intergenic spacer, proposed *P.* subg. *Philodendron* as a paraphyletic group and was unable to solve the (*P.* subg. *Meconostigma* + Asian + American *Homalomena*) polytomy (Fig. 5A). *Gauthier, Barabé & Bruneau (2008)* recovered the American *Homalomena* as sister to *Philodendron* and the Asian *Homalomena* as sister to the (American *Homalomena* + *Philodendron*) clade, although their Bayesian analysis inferred a paraphyletic *Philodendron*, with *P.* subg. *Pteromischum* sister to the American *Homalomena* (Figs. 5B and 5C, respectively). Alternatively, *Cusimano et al. (2011)* recovered a monophyletic *Philodendron*, with *Homalomena* as sister lineage of *Furtadoa* (Fig. 5D). Recently, *Yeng et al. (2013)* estimated the *Homalomena* phylogeny based on the nuclear ITS marker and also sampled *Philodendron* species. In the ML and Bayesian trees reported in their study, *P.* subg. *Pteromischum* was closely related to the

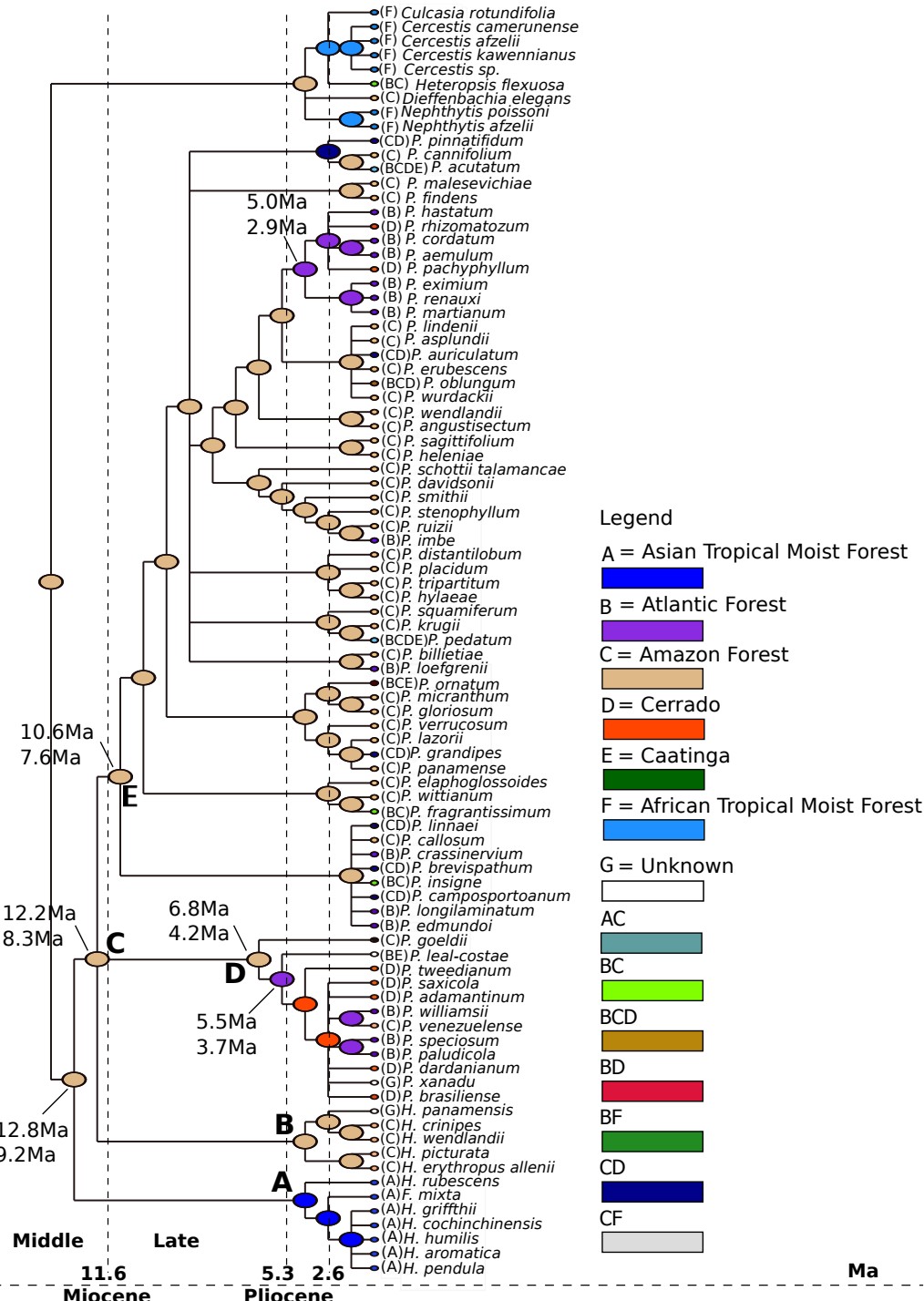

**Figure 4** **Ancestral biome reconstructions and divergence time estimates of *Philodendron* and *Homalomena* lineages.** The epoch intervals followed the international chronostatigraphic chart (*Cohen et al., 2015*) and are indicated by dashed lines.

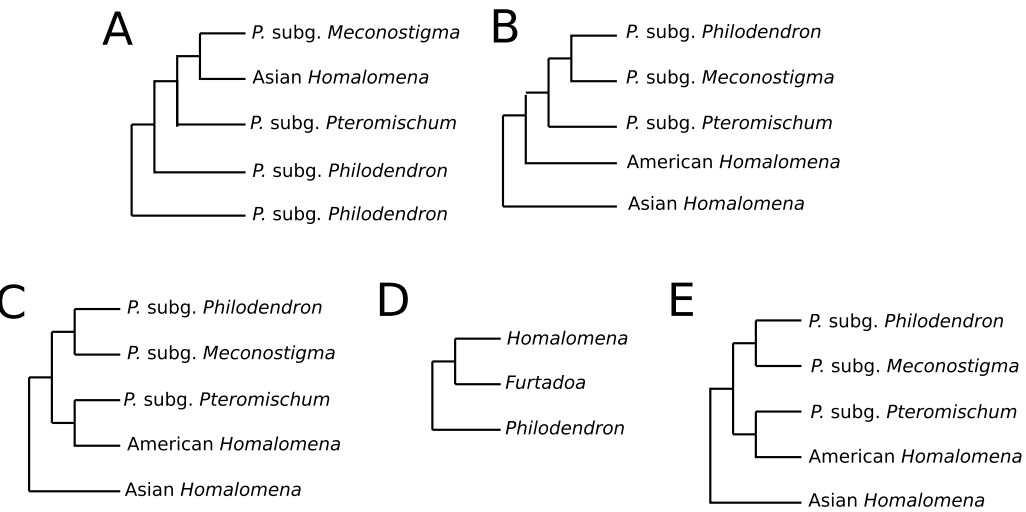

**Figure 5 Phylogenetic relationships between *Philodendron* and *Homalomena* recovered by previous studies.** (A) *Barabé et al. (2002)*; (B) *Gauthier, Barabé & Bruneau (2008)* using the maximum parsimony method; (C) *Gauthier, Barabé & Bruneau (2008)* using Bayesian analysis; (D) *Cusimano et al. (2011)* (2011); (E) *Yeng et al. (2013)*.

American *Homalomena*, whereas *P*. subg. *Meconostigma* and *P*. subg. *Philodendron* were recovered as sister taxa (Fig. 5E).

Discrepancies between previous works and our analysis may be due to different choice of phylogenetic methods, markers and taxon sampling. *Gauthier, Barabé & Bruneau (2008)* was the only study intended to investigate specifically the systematics of *Philodendron* genus. When compared to their analysis, our study included a larger sampling of taxa and molecular markers with the aim of estimating the phylogeny of *Philodendron* and *Homalomena* species; it is also the first analysis that used a supertree approach to this end.

Our phylogeny characteristically presents short branch lengths within the *Philodendron* clade. The high frequency of polytomies indicates the genetic similarity among terminals, which is further corroborated by the ease in obtaining artificial hybrids between different species. This corroborates a scenario of low genetic differentiation and low reproductive isolation (*Carlsen, 2011*).

*Philodendron* diversification may also consist of several recent rapid radiation events. Phylogenetic reconstruction under this scenario is challenging, because of a significant amount of substitutions is needed to accumulate within short periods of time (*Maddison & Knowles, 2006*). However, morphological variation of *Philodendron* is remarkable, which seems contradictory considering the previously discussed features. However, it has been extensively discussed that morphological variation is not a suitable proxy for genetic variation (e.g., *Prud'Homme et al., 2011*; *Houle, Govindaraju & Omholt, 2010*). Many environmental and epigenetic factors may can increase phenotypic variation even in the absence of DNA sequence variation (*Prud'Homme et al., 2011*). Evidently, we cannot rule out the possibility that DNA regions that present significant genetic differences between species were not sampled in this work.

## Diversification of *Philodendron* and *Homalomena*

Although the chronology of *Philodendron* divergence was not extensively focused by previous studies, *Nauheimer, Metzler & Renner (2012)* analyzed the global history of the entire Araceae family based on a supermatrix composed of five chloroplast markers and several well-established calibration points. Their analysis included a single *Philodendron* species and estimated age of the *Philodendron*/Asian *Homalomena* divergence at approximately 20.0 Ma (ranging from 31.0–9.0 Ma). This study, however, also included a single species of Asian *Homalomena*.

The wide range of the posterior distribution credibility intervals of *Nauheimer, Metzler & Renner (2012)* hampers the proposition of putative biogeographic scenarios for the evolution of *Philodendron*, American and Asian *Homalomena*. Differences between their timescale and the divergence times proposed in this study might therefore be due to methodological differences caused by their reduced taxonomic sampling. Nevertheless, both our estimate of the age of the *Philodendron* divergence from Asian *Homalomena* and that of *Nauheimer, Metzler & Renner (2012)* suggests that this event took place when South America was essentially an isolated continent.

The isolation of the South American continent persisted from approximately 130.0 Ma (*Smith & Klicka, 2010*) to 3.5 Ma (*Vilela et al., 2014*), with the rise of the Panamanian land bridge. Therefore, from the Early to Middle Miocene there was no land connection with North America, Asia or Africa (*Oliveira, Molina & Marroig, 2010*). If dispersal, rather then vicariance, is the most plausible hypothesis to explain *Philodendron* and American *Homalomena* colonization of the Neotropics, hypotheses on the possible routes of colonization should be investigated. Based on the continental arrangement during the Miocene, we propose that the dispersal of *Philodendron* and American *Homalomena* ancestor could have followed four possible routes (Fig. 6): (1) from Asia to North America through the Bering Strait; (2) from Africa to the Neotropics by crossing the Atlantic ocean; (3) from Asia to Neotropics by crossing Pacific ocean; and (4) from Asia to Neotropics , also by crossing the Atlantic ocean.

The Araceae fossil record is currently assigned to Florida, Russia, Germany, United Kingdom, Venezuela, Yemen, Colombia and Canada (*Shufeldt, 1917*; *Berry, 1936*; *Bogner, Hoffman & Aulenback, 2005*; *Chandler, 1964*; *Dorofeev, 1963*; *As-Saruri, Whybrow & Collinson, 1999*; *Wilde & Frankenhauser, 1998*; *Wing et al., 2009*; *Stockey, Rothwell & Johnson, 2007*). However, as none of the fossil specimens was described as closely related to *Philodendron* or *Homalomena*, the Araceae fossil record fails to corroborate any dispersal hypothesis in particular.

Considering route 1, although the Bering Strait have connected Asia to the North America during most of the Cenozoic period (*Butzin et al., 2011*), there is no evidence of extant *Philodendron* and *Homalomena* in North America or North Asia. Route 2 involves long-distance oceanic dispersal through ca. 2,000 km—the minimum distance between Africa and the Neotropics (*Oliveira, Molina & Marroig, 2010*)—through Atlantic paleocurrents, which were probably stronger than Pacific currents. This hypothesis is congruent with the clustering of *Philodendron* and American *Homalomena* into a single clade, assuming Africa as the center of diversification of Asian and American *Homalomena*, as well as *Philodendron*.

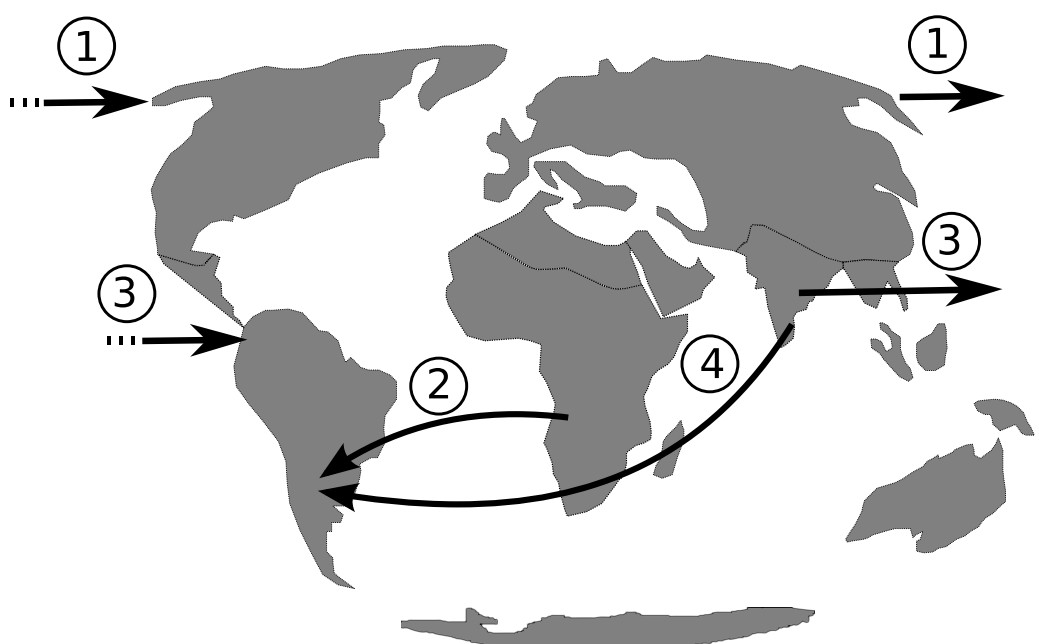

**Figure 6** Putative dispersal routes of the ancestor of *Philodendron* and American *Homalomena* to the Neotropical region during the Miocene.

However, we should conisder that the last recent common ancestor of *Philodendron* and *Homalomena* was distributed in Africa. On the other hand, this hypothesis is corroborated by the distribution of the extant *Philodendron* and *Homalomena* species. Givnish and colleagues (*2004*) also suggested two long-distance dispersal events through the Atlantic, but in the opposite direction. Their analysis indicated that Bromeliaceae and Rapateaceae arose in the Guayana Shield of northern South America and reached tropical west Africa via long-distance dispersal at ca. 6–8 Ma.

When considering long-distance dispersal events, it is crucial to evaluate their viability as related with the plant's ability to produce dispersal structures that would tolerate aquatic and saline conditions for long periods of time (*Lo, Norman & Sun, 2014*). Although such features have not been evualuated for *Philodendron* and *Homalomena*, some *Homalomena* species inhabits swamp forests and open swamps. Thus, features that would favor their survival in waterlogged environments could also influence their maintenance in seawater.

Although route 3 is geographically unlikely due to the 8,000 km distance between Asia and the Neotropics through the Pacific Ocean (*Oliveira, Molina & Marroig, 2010*), it cannot be completely discarded because it is corroborated by the extant distribution of *Homalomena* and *Philodendron*. Finally, route 4 suggests the dispersal through the Atlantic ocean from Asia to the Neotropics. This is also an improbable hypothesis because the African continent would act as a barrier between Asia and the Neotropics, requiring the dispersal through both the Indian and the Atlantic oceans.

The extant distribution of *Philodendron* and *Homalomena* species and the scarcity of fossil information challenge the proposition of a scenario for the origin of *Philodendron* and American *Homalomena* in the Neotropics. However, the biological and geographical
information provided to date indicates a long-distance oceanic dispersal through the Atlantic, as suggested by route 2, as the most plausible hypothesis to explain *Philodendron* and American *Homalomena* colonization of the Neotropics.

### Early diversification of *Philodendron* species

According to our analysis, the last common ancestor of *Philodendron* and the American *Homalomena* was distributed in the Amazon forest about 8.6 Ma (11.1–6.8 Ma) during the Middle/Late Miocene. Interestingly, this time estimate is very close to the age of the divergence between the (*Philodendron*/American *Homalomena*) clade from the Asian *Homalomena* (Fig. 4, node A). The Middle and Late Miocene were characterized by wetland expansion into western Central Amazonia, which fragmented the rainforest and formed extensive wetlands (*Jaramillo et al., 2010*). According to our analysis, *Philodendron* earliest divergence events took place in this scenario. The Amazon forest, from the Late Miocene to the beginning of Pliocene, was composed of a diverse and well-structured forest. The Amazon river landscape was well established; this probably allowed the extensive development of the Amazonian *terra firme* forest (*Jaramillo et al., 2010*). This scenario is compatible with the biology of extant species of *Philodendron* because a well-structured forest would allow the development of epiphyte and hemiepiphyte species, such as *Philodendron*.

### *Philodendron* diversification along Neotropical biomes

Our results suggest that *Philodendron* species occurred exclusively at the Amazon forest for ca. 5.0–6.0 Ma. During the Pliocene, as result of the glacial cycles, climate cooling and drying permitted the expansion of the open savanna areas, mostly represented by the 'dry diagonal', which is constituted by the Caatinga, Cerrado and Chaco biomes. This consisted of a crucial event, because it resulted in the isolation of the Atlantic forest in the east coast of South America (*DaSilva & Pinto-da-Rocha, 2013*), which is synchronous to the inferred age of the early diversification of *Philodendron* in this biome. This also corroborates the hypothesis that the Atlantic forest taxa present a closer biogeographic relationship with the Amazon forest, as proposed by *Amorim & Pires (1996)* and *Eberhard & Bermingham (2005)*. After the separation between Atlantic and Amazon forests during the Pliocene, species dispersal might have been common through the forest patches (*DaSilva & Pinto-da-Rocha, 2013*).

*Roig-Juñent & Coscarón (2001)* and *Porzecanski & Cracraft (2005)* suggested that the Atlantic rainforest also presents similarities in organismal composition with the Cerrado biome. This association would have been a result of dispersal events through gallery forests. The history of the formation of Cerrado biome is still uncertain (*Zanella, 2013*; *Werneck, 2011*), but our analysis indicated that the ancestors of *Philodendron* clades from the Cerrado were distributed in the Atlantic forest. Therefore, we also corroborate the hypothesis of lineage dispersal from the Atlantic forest to the Cerrado biome. These events would have occurred after the colonization the Atlantic forest by *Philodendron* species.

### Final considerations on *Philodendron* evolution

Given the significant morphological diversity of *Philodendron*, its widespread distribution in the Neotropics and the age of the Araceae family (~140.0 Ma, *Nauheimer, Metzler & Renner, 2012*), it would be expected that the origin of this genus was older. In sharp contrast, we have estimated phylogenies with very short branch lengths and very recent divergence times. A similar scenario was reported by *Carlsen & Croat (2013)* for *Anthurium*, which is the most diverse Araceae genus, and also by Nagalingum and colleagues (*2011*) for cycads. Therefore, the inferred tempo and mode of evolution of *Philodendron* species were reported in several plant families.

## CONCLUSION

The present work was the first attempt to establish a chronological background for the diversification of this highly diverse genus and to suggest possible routes of colonization of the ancestors of Neotropical *Philodendron* and *Homalomena*. *Philodendron* was statistically supported as a monophyletic genus, sister to American *Homalomena* by AU and SH tests. The last common ancestor of *Philodendon* diversified from the Middle to the Late Miocene in the Amazon forest, where the earliest events of *Philodendron* diversification occurred. Amazon was also the exclusive biome occupied by *Philodendron* species during a 5.0–6.0 million years period. Atlantic forest lineages of *P*. subg. *Meconostigma* and *P*. subg. *Philodendron* diverged from Amazonian ancestors. On the other hand, the majority of Cerrado species evolved from Atlantic forest ancestors, from the Late Miocene to the Pliocene.

## ACKNOWLEDGEMENTS

We thank Petrobrás and INPA for allowing field expeditions in their biological reserves. We also thank Alexandre Antonelli for valuable contributions in the manuscript text.

### Funding

CGS was funded by the Brazilian Research Council-CNPq grant 307982/2012-2 and the Rio de Janeiro State Science Foundation-FAPERJ grants 110.028/2011 and 111.831/2011. The funders had no role in study design, data collection and analysis, decision to publish, or preparation of the manuscript.

### Grant Disclosures

The following grant information was disclosed by the authors:
Brazilian Research Council-CNPq grant: 307982/2012-2.
Rio de Janeiro State Science Foundation-FAPERJ grants: 110.028/2011, 111.831/2011.

### Competing Interests

The authors declare there are no competing interests.

## Author Contributions

- Leticia Loss-Oliveira conceived and designed the experiments, performed the experiments, analyzed the data, wrote the paper, prepared figures and/or tables, reviewed drafts of the paper.
- Cassia Sakuragui conceived and designed the experiments, analyzed the data, contributed reagents/materials/analysis tools, reviewed drafts of the paper.
- Maria De Lourdes Soares conceived and designed the experiments, contributed reagents/materials/analysis tools.
- Carlos G. Schrago conceived and designed the experiments, performed the experiments, analyzed the data, contributed reagents/materials/analysis tools, wrote the paper, prepared figures and/or tables, reviewed drafts of the paper.

## DNA Deposition

The following information was supplied regarding the deposition of DNA sequences:
GenBank accession numbers are listed in Tables S1 and S2.

## Supplemental Information

Supplemental information for this article can be found online at http://dx.doi.org/10.7717/peerj.1744#supplemental-information.

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
