# Peer review of "Evolution of Philodendron (Araceae) species in Neotropical biomes"

_PeerJ, doi:10.7717/peerj.1744_

## Round 0.1 · original submission · Major Revisions

Both reviewers appreciated the contribution that your paper makes. Well done! However, they also identify several shortcomings and many minor issues that require attention before I can accept your paper. I agree with Reviewer 1 that the focus of the paper is on historical biogeography and systematics, rather than on diversification, as claimed in the title. Reviewer 1 has many other pertinent comments.

·

Basic reporting

1. English is not always clear and sometime with unambiguous text
2. Objectives can be more precisely defined. e.g. as suggested by the title, the submission attempts to infer rapid diversification and evolution in Philodendron, in fact, much of the content focused on ancient range reconstruction, the inference of the evolutionary relationships between Philodendron and Homalomena.

Finally, this submission reported a few results inconsistent with previous reports, other than attributing the inconsistence of this study with previous results as “methodological differences”, or “sampling issue”, it is worth including some thorough discussion to explore the biological mechanisms behind these inconsistence, and how this current study has provided new understanding into those possible biological mechanisms.

Experimental design

The numbers of taxa analysed, as described in the method section, are inconsistent with what was presented in the abstract and in the figure, and again inconsistent with the number of species in supplement table. clarifications are needed to explain these inconsistence.

Meanwhile, it is worth presenting more details on how the sampling regime aimed to cover maximum taxonomic diversity and diverse geographic distribution (if that was the case), and providing some consideration on whether the incomplete sampling would have impacted on main result and conclusion, given that there are 489 species while only 68 were finally included in the analysis.

Validity of the findings

Inference of phylogenetic tree needs to be more rigorous, and with more details presented. The method of Inference of phylogenetic tree could be better (see below specific comments). Information to describe, e.g. phylogenetic uncertainty, etc. should be included. e.g. “wide range posterior distribution credibility” was proposed to prevent proposition of putative biogeographic scenarios for the evolution of Philodendron” in previous study, while this submission even did not report wide posterior credibility distribution at all. Such information is very important for the readers to know how confident the conclusion could be drawn.

L139-155. The approach of constructing phylogenetic tree (supertree, as called in the manuscript) seems wasted a considerable proportion of information, as over a third of sequenced species were removed from further analysis due to incongruence between gene trees. In phylogenetic study with multiple loci, combining data sets is a good approach. it is important to have congruent trees and not simply select the best trees and go with them throwing away the rest. If the genes trees are incongruent, there is even more reason to combine them, since it is only through multi-loci datasets so that gene-specific bias can be minimized, allowing true phylogenetic signal drive the tree. Both MrBayes and BAEST allow you to construct phylogenetic tree with concatenated sequences with appropriate model setting.

Additional comments

L1: No diversification was actually studied, e.g. no diversification rate was actually reported. a more precise title to reflect this fact is needed.

L70: Richness of what?

L83. It is not wrong to say “more than 300 species”, but it can be more precise. You have 489 formally recognized species in the genus, 21 species in subgenus Meconostigma; 75 in Pteromischum, there are close to 400 species in subgenus Philodendron.

L96-99. This sentence is a bit confusing. Perhaps write as: “In accordance to previous analysis (e.g. Mayo et al., 1997; Barabé et al., 2002), the authors reported that Philodendron is not a monophyletic group, with American species from morphologically similar genus, Homalomena Schott, embodied within Philodendron clade”?

L104. “a single colonization event…”, this is purely speculative, and is unnecessary information. Consider remove.

L141. Not “Mr.Bayes”. it is “MrBayes”.

L207-208. The supertree estimated in PhySIC_IST was not time-based, as described in the methods section. A few issues need to be classified. 1), the figure presenting ancestral range re-construction has clear time scale, whether this was the same tree as the supertree that was described as the input tree? 2), In “Divergence time inference”, the authors used the evolutionary rates of plastid coding regions of monocots. Nuclear genes were excluded…”, while the input Philodendron tree (supertree?) for analysis in RASP was estimated from a combination of plastid and nuclear genes.

L217-228 It would be helpful to present a phylogenetic tree with clear indication of support at each node, at least for those key nodes.

L230. Was “6.8-11.1Ma” 95% HPD?

L230. But Node A in Fig 4 has been indicated as “9.2Ma”. Note B?

Fig 4: the majority of outgroup specie are Asian tropical species, and basal species (Asian Homalomena) are Asian origin, logically, one would expect the root of the entire tree would be asian origin, Node A would be Asia, other than Amazon. By the way of it currently present, figure suggested a dispersal from Amazon to Asia, not as claimed in the discussion (L ) and the in last figure 6

L231. Middle and Late Miocene would cover a period of time from ca 13.8 Ma to 5.3Ma. in Figure 4, What do the broken lines and the number at the bottom of the lines mean? They are time points to separate middle (Miocene?) and late Miocene? If so, 11.0, 7.0 and 2.6 are arbitrary. The line with 11.0 at the bottom cross the tree between note b (8.6ma) and note c (7.6ma), the same inconsistence are for the other two broken lines.

L257. “Therefore, the monophyly of Philodendron remains unsolved”, but in the result (L223-224), it is said “the topological arrangement in which Philodendron is monophyletic genus was significantly supported by the AU and SH tests”.

L276-282. Given the mixed results on the evolutionary relation of Philodendron and American Homalomena, it would be more interesting to include some discussion on possible biological mechanism that cause such inconsistence.

L308-312: it is unnecessary to propose route 2 and 3 at all since their complete impossibility.

L315: “No extant species or fossil record in North America” is not a very strong reason to reject the hypothesis, because that doesn’t necessary mean there were no species there 11 or so million years ago.

L300-340. So which one is the most likely scenario?

·

Basic reporting

The language is not particularly bad, but mistakes are still fairly numerous, and so the manuscript would have to be read by a native English speaker prior to publication. Common mistakes seem to be verbs that should be plural in the singular and vice-versa, and use of wrong prepositions (across/along/through).

Experimental design

The authors have sampled a wide array of species, and the science behind the paper appears solid.

Validity of the findings

The findings are interesting and largely realistic, with relevance for taxonomy, biogeography and biome-level distributional history. The movement out of rainforest into other biomes is a particularly interesting confirmation of what is known on the antiquity of these biomes.

Additional comments

Suggestions:

Some parts of the results section may belong better in the discussion.

The discussion could use a bit more structuring. I am not sure whether all four biogeographic hypotheses deserve to be mentioned. It may be good to start with a description of the fossil record for the family, then move to any fossils in the clade relevant here, talk about means of dispersal (seeds but also any asexual reproduction) and then build on that with relevant hypotheses. The conclusion could try to be a bit more general. There is room for comparing results with other studies than the two cited there, but that would be another section to the discussion rather than a conclusion. In comparing Atlantic crossings, it would be worth mentioning Rapateceae and Bromeliaceae, both of which went in the opposite direction.

I would personally prefer to see slightly more intuitive representations in figures 1, 4 and 6. Figure 1 could have rainforest, cerrado and caatinga portions of the range in different colours or shading. Figure 4 could use different means of indicating biome and region – maybe stick to the ovals currently in use for rainforest only and colours for regions, and use different shapes for different biomes. This could be aligned with the revised Figure 1. Figure 6 could use some improvement in the way the arrows are presented for clarity.

Minor points:
Line 317: Araceae not Areceae (a tribe in Arecaceae).
The first reference in the list starts with a year.

---

## Round 0.2 · accepted · Accept

I am satisfied with the detailed responses to two demanding reviews. I suggest one very minor change to the title: use 'in' instead of 'along'. Thank you for this useful contribution.